# Regional variations in child health deprivation and its associated factors in Nigeria

Victor Chima[1], Funmilola F. Oyinlola[1], Joseph A. Kupoluyi[1]*, Segun Tekun[2], Ifeyinwa U. Anyanyo[3]

**1** Department of Demography and Social Statistics, Faculty of Social Sciences, Obafemi Awolowo University (OAU), Ile-Ife, Nigeria, **2** Centre for Sustainable Development, University of Abuja, Abuja, Nigeria, **3** Department of Public Health, Faculty of Health Sciences, Texila American University, Guyana, Georgetown, Guyana

* jakupoluyi@oauife.edu.ng

## Abstract

Child health deprivations differ by socio-cultural differences and some demographic and socio-economic factors. Deprivation may be more pronounced by the geo-political zones/regions in Nigeria given the differences in their socio-cultural, education, religion, and economic particularly between the North and the South geo-political zones. Thus, this study examined regional variations in child health deprivation and its associated factors in Nigeria. The 2021 Nigeria Multiple Indicator Cluster Survey (MICS) was used for the study. A weighted sample size of 26,639 under-five children was analysed using STATA SE Version 14. Data were analysed using different descriptive statistics to examine regional variations in child health deprivation. Pearson's Chi-square and Binary logistic regression were performed to determine associated factors influencing child health deprivation in Nigeria at $p < 0.05$ level of significance. Results showed that nearly all children (96%) experience at least a deprivation in healthcare with a slight variation across regions in Nigeria. Child health deprivation was higher in the Northern regions than in the Southern regions. Interestingly, when compared to those who were not deprived, the Southwest region had the highest percentage of children who were not deprived (5%). The study also found lower odds of child health deprivation between children aged four (4) (OR = 0.65, 95%CI [0.50-0.85], $p < 0.05$), whose household head had tertiary education (OR = 0.19, 95%CI [0.13-0.28], $p < 0.05$), and from richest wealth index (OR = 0.06, 95%CI [0.04-0.10], $p < 0.05$). The study concludes that health deprivation is high among children in Nigeria irrespective of region of residence. Household and other factors have effects on the deprivation of healthcare for children according to the region of residence. This accentuates the need for a comprehensive review of policies and strategies related to health insurance schemes, and vaccination programs targeting under-five children in Nigeria.

**Data availability statement:** The datasets are available online at https://mics.unicef.org/surveys.

**Funding:** The authors received no specific funding for this work.

**Competing interests:** The authors have declared that no competing interests exist.

## Background

Universally, children are identified as the most vulnerable in society and are remarkably affected by poverty, particularly in the age range of 0 – 15 years [1]. Child poverty is one of the defining issues faced globally as more under-five children are reported to be living in poverty. The multidimensional poverty report of 2022 reveals that 1.2 billion people in 111 countries live in acute multidimensional poverty, with almost one-third of these children deprived of education, health, sanitation, housing or nutrition and of opportunities and dignity [2]. Although there seems to be a reduction in the proportion of children living in extreme poverty according to the report in the last ten years (a fall from 50.2% to 43.4%), children are still observed to be more than twice as likely as adults to live in extreme poverty [3]. This indicates the need to investigate the hazardous situation affecting the future generation of the global community [3].

Children experience deprivation and poverty differently when compared to adults, and this ultimately affects their development [4]. Child Fund International (CFI) defined child deprivation using three keywords; deprivation, exclusion, and vulnerability. Deprivation is the lack of material services and conditions for under-five children. Exclusion, in its own right, refers to the denial of safety and rights of under-five children due to the incapacity they have to be independent of themselves, and Vulnerability is a situation that occurs when society cannot address the threats faced by children, especially under-five children [5].

Generally, child deprivation is measured using different indicators in nutrition and food security, healthcare, water, sanitation facilities, shelter, education, information, and protection [6]. As defined by UNICEF using the multidimensional approach, child health deprivation connotes the experience of a lack of access to nutrition, water and sanitation facilities, basic healthcare services, shelter, education, participation, and protection by children. Particularly, under-five children are characterized by deprivation experience in the areas of development/stunting, nutrition, health, water, sanitation, and housing. Development/stunting is measured in terms of the height of the child's height which is below -2SD from the reference population; child health deprivation is indicated in incomplete vaccination, lack of knowledge of mothers in the treatment of diarrhoea, and limited ANC visits by mothers; deprivation in nutrition is showcased when the child is not exclusively breastfed, the child is not fed with the Minimum Acceptable Diet (MAD), the child did not receive a Vitamin A supplement during the last 6 months, etc.; water deprivation is experienced when the household in which the under-five child lives in uses an unimproved water source; sanitation deprivation is indicated when the under-five child lives in a household that does not have access to improved toilet type; and housing deprivation is experienced by under-five children when such children are exposed to indoor pollution by the usage of solid cooking fuels for cooking within the house without a designated place as a kitchen [7].

Health is a basic dimension of well-being and an important component of human capital. A strategic dimension of child health deprivation is determined by poor health and the incapacity to cope with a series of ailments [6]. Under-five

child health deprivation remains a pressing concern in Nigeria, with alarming disparities evident across regions. According to UNICEF (2024), Nigeria accounts for approximately 10% of global under-five deaths, with an estimated 746,000 deaths annually. The country's under-five mortality rate stands at 118 deaths per 1,000 live births, significantly higher than the global average of 39 deaths per 1,000 live births [8]. Stunting affects 43.6% of Nigerian children under five, exceeding the global average of 22% [9]. Regional variations are stark, with under-five mortality rates ranging from 104 deaths per 1,000 in the South-West to 159 deaths per 1,000 in the North-West [10]. Immunization coverage for basic vaccines also lags, at 54% versus the global average of 85% [9]. This challenge is similar to other regions of the world. As reported, South- Asia's under-five mortality rate, although declining, remains high at 43 deaths per 1,000 live births, with India and Pakistan bearing the largest burdens [9]. Latin America's under-five mortality rate averages 14 deaths per 1,000 live births, but countries like Haiti and Bolivia lag [8,9]. In sub-Saharan Africa, countries like the Democratic Republic of Congo, and Ethiopia contribute significantly to under-five mortality rates, exceeding 100 deaths per 1,000 live births [9].

Globally, the healthcare system faces daunting challenges, including unequal access, inadequate funding, and disparities in quality care [9]. Despite progress, millions lack essential health services, with approximately 800 million people spending at least 10% of their household budgets on healthcare [11]. Achieving universal health coverage (UHC) remains an urgent priority, requiring collaborative efforts to strengthen health systems, innovate financing models, and address social determinants of health [12]. Similarly, the healthcare system in Nigeria is subjected to challenges such as poor governance, dilapidated healthcare facilities, human resource challenges, and financial constraints, which affect the level of access and distribution of healthcare services by the population, especially, those under-five children [8].

Child health deprivations are identified to be associated with several recognizable causal factors [1]. In Nigeria, child health deprivation is linked to inequalities in income and wealth gaps in many households. Children from poor households are more susceptible to health deprivation and lack of access to quality health care [13]. Further, quality healthcare for children is significantly determined by the characteristics of the family including parenting practices, parental behaviour, family structure and parent socio-economic status [14]. Additionally, health deprivations, especially in rural regions are linked to household size, age of household head, sex of household head, and household economic activities [1]. Parental education also significantly impacts child health, as educated mothers are more likely to seek prenatal care, immunize their children, and adopt healthy practices [14]. Access to maternal health services also plays a critical role, with studies showing that skilled birth attendance reduces child mortality rates [15]. However, cultural beliefs and practices can hinder healthcare-seeking behaviour, particularly in communities where traditional remedies are preferred [15]. Regional healthcare infrastructure disparities further exacerbate child health deprivation, with rural areas often lacking access to quality care [11]. Additionally, gender inequality perpetuates health disparities, as girls' health needs are often neglected, and women's decision-making power over healthcare is limited [16]. While these factors largely determine the health status of a child, there is a disparity in the prevalence of child health deprivation across regions and areas, and across the different socio-economic levels in Nigeria [11].

Africa has the highest incidence of child health deprivation [8]. This is exacerbated by the increasing rate of urbanization which brings disparity in the level of access to health services, as only the urban rich can easily access health services. In countries like Angola, Senegal, and Central African Republic, the experiences of child health deprivation by the urban poor are not entirely different from those living in rural areas. Thus, child health deprivation is experienced equally in the rural and urban areas, especially in the slum areas of the urban centres [17]. Child poverty is reflected both in rural and urban areas, and under-five children in rural areas are found to be deprived of certain beneficial and advantaged resources such as healthcare [4]. Pointedly, inequality in access to basic social amenities also exists in both rural and urban centers; the rate of migration of people from rural to urban centers affects the distribution and access to resources in both centers, however, a large chunk of the consequences is depleted in the rural centers, with the under-five children

largely being on the receiving end [18]. Also, the general belief has been that there is wide access to basic amenities like water, sanitation, hygiene, etc. in urban areas, however, it has been proven by studies that dwellers in urban areas, particularly the slum areas often face remarkable challenges as being faced by rural dwellers. These challenges lead to health risks such as diarrhoea, cholera, malaria, etc. which are mostly borne by under-five children [19]. In Nigeria, two-thirds of children under-five do not receive advice or treatment for diarrhoea from a health facility or provider, and about 40 percent of under-five children who have a fever do not get treatment from a health facility or provider. These health deprivations differ by region and are influenced by some demographic and socio-economic factors. Furthermore, 64 per cent of children are deprived of immunization as almost 20 percent of these did not receive any immunizations while others did not receive complete doses. Deprivations in immunizations differ across geo-political regions as more children are deprived of immunizations in the Northern region when compared to the Southern region. In the three Northern geo-political zones, at least 65 per cent of the children are deprived of complete immunizations while the proportion is at most 51 per cent in the Southern geo-political zones [1].

Studies on regional variation of child health deprivation and its associated factors in Nigeria remain scarce and limited. Previous studies [5,13,14,18–20] focused on poverty, child poverty, and its associated factors with child health outcomes. These studies somewhat focused on single indicators to measure child health deprivations or limited in its approach to examine regional variations by associated factors, hence, this study aims to investigate regional variations in child health deprivation and its associated factors in Nigeria. This study uncovers the intricate factors contributing to child health deprivation in Nigeria, providing valuable insights for data-driven policies and interventions aimed at reducing mortality rates among children under-five. By pinpointing regional disparities and socioeconomic determinants, the research informs strategic investments in healthcare infrastructure, education, and community engagement, ultimately paving the way for improved health and well-being of Nigeria's most vulnerable children. It also addresses the sustainable development Goal 10- reduced inequalities in Nigeria.

## Rationale for investigating regional variations

The study hinged on the Cosmopolitan-Success and Conservative-Failure Hypothesis (CSCFH) [21]. The hypothesis was extended to explain regional differences in Nigeria [22]. The hypothesis suggests that societies that embrace diversity and are open to change are more likely to experience success, while societies that resist change may face challenges in achieving their goals. The hypothesis stipulates that more cosmopolitan societies (characterised by a willingness to embrace new ideas, technologies, social changes, and cultural practices) are more likely to experience greater success in various aspects of life, including health, education, social programmes and interventions, while the more conservative societies (characterized by a strong adherence to promote traditional institutions, customs and values, and resistance to change in the socio-cultural, political philosophy and ideology) may face sluggish or little progress or failure, particularly when it comes to adopting new ideas or innovations that are perceived to contradicting some aspects of the traditional culture.

Several studies [21–25] have used CSCFH in explaining regional variations in Nigeria. The Southern part of the country is considered more cosmopolitan, while the Northern part, is more conservative, leading to differences in outcomes related to education, healthcare, and social matters. The southern part is often associated with higher levels of education, wealth, vaccinations, and adoption of modern healthcare practices, while the Northern part is associated with early childbearing, poor healthcare utilization, and higher rates of child morbidity and mortality among other health indicators. Furthermore, the Northern part of the country suffers from poor immunization services, fever treatment, health insurance schemes, severe environmental pollution, and limited access to clean water, coupled with poor sanitation which increases the prevalence of diarrheal diseases, and malnutrition.

Nigeria as a country was formed through the amalgamation of the Northern and Southern protectorates by Lord Frederick Luggard in 1914 for easy administrative purposes. The country has 36 states and a Federal Capital Territory (Abuja).

Administratively, with more than 250 ethnic groups among which Yoruba, Hausa/Fulani and the Igbo are the dominant groups, Nigeria was grouped into six geopolitical zones/regions and 774 constitutionally recognised local government areas (LGAs) [25]. Largely, the six geopolitical zones are categorized into two: North (comprising North Central, North East, and North West) and South (South East, South South, and South West).

Applying the principle of the theory to the study, it was hypothesized that the Northern children will experience health deprivation than the Southern part and that efforts by the government and non-governmental in health programmes and interventions will be more successful in the Southern part of the country than in the Northern part. Hence, motivated by these observed disparities, the present study seeks to examine regional variations in child health deprivation and associated factors in Nigeria.

## Methodology

### Data source

Data from the Multiple Indicator Cluster Survey (MICS) 2021 (Secondary data), a nationally representative household sample survey of 38,806 households residing in non-institutional dwelling units in rural and urban Nigeria were used for this study. The survey provided estimates of the key health indicators at the national and regional levels, as well as for urban and rural areas. A total of 31,103 under-five children were interviewed for the survey. Thus, in this study, the data of children were obtained from the MICS datasets and analyzed.

### Survey design

The MICS 2021 employed a cross-sectional research design. The survey utilized a multi-stage stratified cluster sampling for the selection of the survey sample. States were identified as the primary sampling units (PSUs) which were considered clusters based on Enumeration Areas (EAs) from the 2006 census EA frames updated in February 2021 [16,17]. The sample of the households was selected in two stages. Each stratum had a specified number of census enumeration areas selected systematically with probability proportion to size while about 20 households were systematically sampled at the second stage. Within each household, all women aged 15–49 years, all under-five children, and all men aged 15 and above, are included in the survey. A random systematic sampling procedure was used in selecting both the enumeration areas and the households for the survey. Out of 31,103 children in the original dataset, a weighted sample size of 26,639 under-five children was utilized after missing values and invalid responses were excluded from the analysis which formed the basis for reporting the study findings. A more detailed explanation of the sample design of the MICS 2021 has been published before and is freely available at https://mics.unicef.org/surveys.

### Operationalization of variables

**Outcome variable.** Four variables were considered for operationalization of child health deprivation; (i) third dose of pentavalent vaccine (penta 3); (ii) care seeking at a health facility or provider for diarrhea; (iii) care seeking at a health facility or provider for fever; (iv) and health insurance coverage (children under-five). Response of receiving the service for each question was coded as '1= Yes' and '0 = No' otherwise. To generate a single, representative value for the outcome variable, deprivation for under-five children was measured as a composite of these four variables with under-five children who did not receive these services considered as healthy deprived and scores as '1' and '0' otherwise. Hence, a child is health deprived and scored as '1' if; (i) the child did not receive the third dose of pentavalent vaccine, (ii) the child did not receive treatment or advice for diarrhoea from a health facility or provider, (iii) the child did not receive treatment or advice for fever, and (iv) the child did not covered by any health insurance and '0' otherwise. Then, we summed the resulting scores of all four indicators ranging from 0 to 4. Therefore, a child is considered to be health deprived if two or more of the conditions stated are achieved. Thus, those who were not covered by full immunization, did not receive diarrhoea

treatment, or fever treatment, and were not covered by the health insurance scheme were coded as '1' and '0' otherwise (i.e., those who were covered by immunization, have received diarrhoea treatment from the health facility, received fever treatment, and covered by the health insurance scheme) [18,19]. A Cronbach's alphas of 0.80 reliability analysis on the composite implies that the combined items have acceptable reliability.

**Explanatory variables.** The geopolitical zones were the main explanatory variable, and it was measured as zones: North Central, North East, North West, South East, South-South, and South West. The other explanatory variables examined were the age of the child, child sex, household head's age and sex, educational level of the household head, household size, number of children in the household, wealth index, and place of residence. The other intervening variables utilized were exposure to media, orphanage status, and the relationship of the child to the household head. All of which was established in previous literature as influencing child health deprivation [18–20].

## Statistical analyses

Data were analysed using both descriptive and inferential statistics. The socio-demographic characteristics of the respondents (under-five children) were examined using descriptive statistics to gain an understanding of the characteristics of the study population, to organize and check the frequency distribution of the respondents respectively, and which was subject to the weight as stated above. Graphical illustrations of dependent categorical variables were presented with pie charts and multiple bar graphs. A bivariable level of analysis between the outcome and independent variables (individual, household, and other factors) of interest was conducted to determine the level of association using inferential statistics. The Chi-square test of association was employed to determine the association between individual, household, and other factors and the level of child health deprivation. However, because differences in the measures of utilization and child health deprivation may be affected by all the factors combined, a multivariable analysis was conducted. The outcome variable, child health deprivation was dichotomous thus, the multivariable analysis was based on Binary Logistic Regression techniques utilized bearing in mind the significant variables at the bivariate level using $p < 0.05$ level of significance. A Multicollinearity test was performed using a variance inflation factor (VIF < 5). All explanatory variables with evidence of no collinearity (mean VIF = 1.34, maximum = 1.85 and minimum VIF = 1.19) were retained in the models while those with evidence of collinearity (VIF > 5) were excluded from the models. Additionally, while examining trends regionally, the geo-political zone was separated at each level and the same statistical process identified above was conducted. This was done to understand the prevalence of the outcome variable and its association with the identified factors on an individual regional basis. Unless otherwise stated, only differences that were significant at the level of 0.05 (in a two-tailed test) were discussed.

Data were analysed using Stata SE Version 14, a statistical analysis program that incorporates the complex survey design. In the MICS, since complex survey design effects are essential in estimating the precision of survey estimates, the household's sampling weight was used in analysing the under-five data. Variances were estimated using the Taylor Linearized variance estimation approach in a manner that reflects the complex survey design. In incorporating the design features into Stata for the analysis, enumeration areas were used as our Primary Sampling Unit (PSU) or cluster. Stratification was done with regions (rural and urban) as stratum identifiers.

## Handling of missing values

Missing values and unknown/invalid responses were excluded from the analysis.

## Ethical consideration

The permission for usage of all the MICS datasets was sought and granted by UNICEF MICS. Being secondary data, details of other ethical considerations are available free at https://mics.unicef.org.

## Results

### Background characteristics of the respondents

Table 1 shows that about one-fifth (21.8%) of the children were of age 4 years. The result also shows an almost equal percentage for child's sex (51% for males and 49% for females). The majority of respondents' household heads were males (92.7%) while 38.4% of them were aged 35–44 years. About half of the household heads (49.6%) had a secondary level of education and above while about two-fifths (39.7%) of the respondents had household sizes between 6–9 members. Also, more than two-thirds (68.6%) of the respondents had between 0–3 children. Nearly a third (32.3%) of the respondents were from the Northwest region zone, while respondent's household wealth index ranges between the poorest (24%) and the richest (16%) with 63.5% residing in the rural areas of the nation.

### Prevalence of types of child health deprivation

The results in Fig 1 show the prevalence of the types of child health deprivation. The results show that the majority of the children (98%) were not covered by health insurance which implies they are deprived. Also, about 35% of the children did not receive full immunization. On the level of diarrhoea treatment among children, about 34% did not receive treatment for diarrhoea while 66% of the children received treatment for diarrhoea. In addition, about 31% did not receive treatment for fever while about 69% received treatment for fever. In all, the result shows a higher percentage of child health deprivation (97%).

### Child health deprivation by region of residence

Fig 2 result shows that child health deprivation is common in all the regions, especially in the Northern regions (North East, North West, and North Central) which is 98% or above. Similar higher levels were also observed in the Southern region, ranging from 95% and 96%. Interestingly, when compared to those who were not deprived, the Southwest region had the highest percentage of children who were not deprived (5%).

### Bivariable analysis

Table 2 shows the bivariable analysis result of the factors associated with child health deprivation in Nigeria. The analysis reveals child's age, the age of the household's head, education of the household head, educational level, household size, number of children in the household, wealth index, geopolitical zone, and the place of residence were all significantly associated with the child health deprivation (p < 0.05). Other factors such as the level of exposure of the household to media and the relationship of the child to the household head were also significantly associated with child health deprivation (p < 0.05). However, factors such as the child's sex, the sex of the household head, and the child's orphanage status were not significantly associated with the child experiencing health deprivation.

### Regional variations in child health deprivation

Table 3 shows regional variation in factors associated with child health deprivation. Results from the North Central geopolitical zone showed that the age of the household head and educational level of the household head, household size, number of children in the household, wealth index, place of residence and exposure to media were significantly associated with the level of child health deprivation (p < 0.05). Similarly, results from the Northeast geopolitical zone revealed the same pattern of association. The results show that the sex of the household head, the level of education of the household head, wealth index, and place of residence were significantly associated with the level of child health deprivation (p < 0.05). In the Northwest region, similar but different results were observed. The result shows that household size, wealth index, place of residence and exposure to media were significantly associated with the level of child health deprivation (p < 0.05). In addition, in the Southeast region, the level of education of the household head, wealth index, and place of residence remains cogent

**PLOS** | **Global Public Health**

**Table 1. Percentage distribution of respondents by socio-demographics characteristics.**

| Variables | Frequency | Percentage |
|---|---|---|
| **Childs Age** | | |
| 0 | 5,089 | 19.1 |
| 1 | 4,860 | 18.2 |
| 2 | 5,309 | 19.9 |
| 3 | 5,563 | 20.9 |
| 4 | 5,819 | 21.8 |
| **Child Sex** | | |
| Male | 13,482 | 50.6 |
| Female | 13,157 | 49.4 |
| **Education of Household Head** | | |
| None | 8,659 | 32.5 |
| Primary | 4,779 | 17.9 |
| Secondary | 8,580 | 32.2 |
| Tertiary | 4,622 | 17.4 |
| **Sex of Household Head** | | |
| Male | 24,687 | 92.7 |
| Female | 1,952 | 7.3 |
| **Age of Household's Head (mean, ±SD)** | **49.1 (±13.5)** | |
| 15–24 | 421 | 1.6 |
| 25–34 | 5,623 | 21.1 |
| 35–44 | 10,222 | 38.4 |
| 45–59 | 7,330 | 27.5 |
| 60+ | 3,043 | 11.4 |
| **Household Size (mean, ±SD)** | **6.6 (±3.3SD)** | |
| 1–5 | 10,057 | 37.8 |
| 6–9 | 10,577 | 39.7 |
| 10+ | 6,006 | 22.6 |
| **Number of Children (mean, ±SD)** | **4.5 (±2.6)** | |
| 0–3 | 18,267 | 68.6 |
| 4–6 | 6,473 | 24.3 |
| 7 and above | 1,899 | 7.1 |
| **Wealth Index** | | |
| Poorest | 6,490 | 24.4 |
| Second | 6,008 | 22.6 |
| Middle | 5,171 | 19.4 |
| Fourth | 4,640 | 17.4 |
| Richest | 4,330 | 16.3 |
| **Geopolitical Zone** | | |
| North Central | 3,820 | 14.3 |
| North East | 4,459 | 16.7 |
| North West | 8,598 | 32.3 |
| South East | 2,379 | 8.9 |
| South South | 2,999 | 11.3 |
| South West | 4,385 | 16.5 |

*(Continued)*

**Table 1.** (Continued)

| Variables | Frequency | Percentage |
|---|---|---|
| **Place of Residence** | | |
| Urban | 9,720 | 36.5 |
| Rural | 16,919 | 63.5 |
| **Total** | **26,639** | **100.0** |

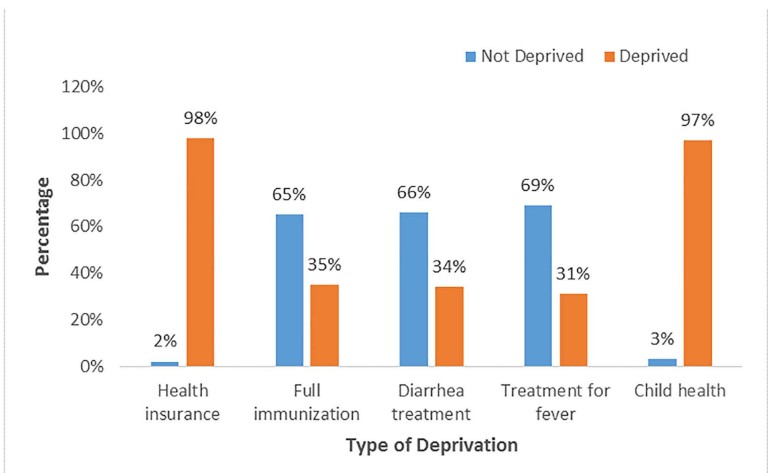

**Fig 1. Prevalence of types of child health deprivation.**

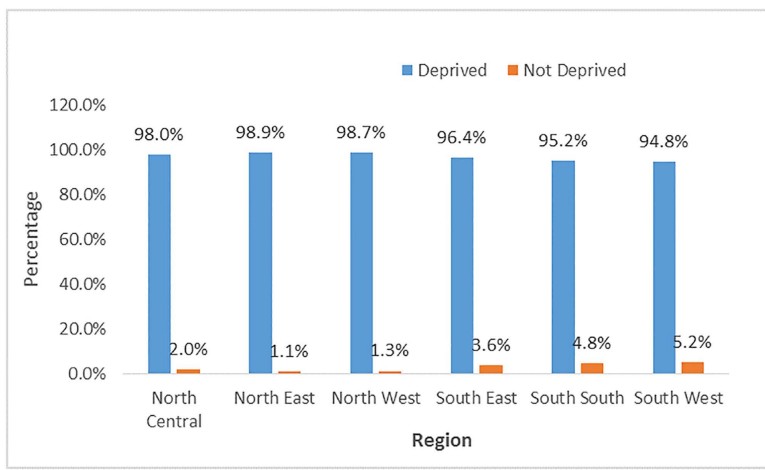

**Fig 2. Prevalence of child health deprivation by region.**

amidst the factors identified as significantly associated with child health deprivation ($p < 0.05$). In the South-South region, the level of education of the household head, wealth index, place of residence, level of exposure to media, and the relationship of the child to the household head were significantly associated with the level of child health deprivation ($p < 0.05$). Also in the Southwest region, the result showed that the sex and age of the household head and the level of education of the

**Table 2. Bivariate analysis of the factors associated with child health deprivation.**

| Factors | Child not deprived (%) | Child Deprived (%) | Chi-square | p-value |
|---|---|---|---|---|
| **Individual Factors** | | | | |
| **Child Sex** | | | | |
| Male | 338 (2.2) | 15,317 (97.8) | 0.024 | 0.876 |
| Female | 331 (2.2) | 14,818 (97.8) | | |
| **Childs Age** | | | | |
| 0 | 99 (1.7) | 5,585 (98.3) | 10.992 | 0.027 |
| 1 | 119 (2.1) | 5,475 (97.9) | | |
| 2 | 148 (2.4) | 6,002 (97.6) | | |
| 3 | 129 (2.0) | 6,295 (98.0) | | |
| 4 | 174 (2.5) | 7,778 (97.5) | | |
| **Household Factors** | | | | |
| **Age of Household Head** | | | | |
| 15–24 | 06 (1.2) | 514 (98.8) | 69.851 | 0.000 |
| 25–34 | 98 (1.5) | 6,495 (98.5) | | |
| 35–44 | 347 (3.0) | 11,099 (97.0) | | |
| 45–59 | 167 (2.0) | 8,373 (98.0) | | |
| 60+ | 51 (1.4) | 3,654 (98.6) | | |
| **Sex of Household Head** | | | | |
| Male | 621 (2.2) | 27.899 (97.8) | 0.057 | 0.811 |
| Female | 48 (2.1) | 2,236 (97.9) | | |
| **Education of Household Head** | | | | |
| None | 39 (0.4) | 10,713 (99.6) | 0.001 | 0.000 |
| Primary | 49 (0.9) | 5,514 (99.1) | | |
| Secondary | 177 (18) | 9,435 (98.2) | | |
| Tertiary | 404 (8.3) | 4,473 (91.7) | | |
| **Household Size** | | | | |
| 1–5 | 331 (3.0) | 10,712 (97.0) | 62.34 | 0.000 |
| 6–9 | 238 (1.9) | 12,138 (98.1) | | |
| 10+ | 100 (1.4) | 7,277 (98.6) | | |
| **Number of Children** | | | | |
| 0–3 | 539 (2.5) | 20,696 (97.5) | 43.66 | 0.000 |
| 4–6 | 105 (1.4) | 7,331 (98.6) | | |
| 7 and above | 25 (1.2) | 2,108 (98.8) | | |
| **Wealth Index** | | | | |
| Poorest | 23 (0.3) | 8,487 (99.7) | 0.002 | 0.000 |
| Second | 27 (0.4) | 7,487 (99.6) | | |
| Middle | 83 (1.3) | 6,418 (98.7) | | |
| Fourth | 163 (3.3) | 4,728 (96.7) | | |
| Richest | 373 (11.0) | 3,015 (89.0) | | |
| **Geopolitical zone** | | | | |
| North Central | 177 (2.8) | 6,159 (9.2) | 143.561 | 0.000 |
| North East | 154 (2.1) | 7,354 (97.9) | | |
| North West | 81 (1.1) | 7,662 (98.9) | | |
| South East | 22 (0.8) | 2,606 (99.2) | | |
| South South | 105 (3.3) | 3,125 (96.8) | | |
| South West | 130 (3.9) | 3,229 (96.1) | | |

*(Continued)*

**Table 2.** (Continued)

| Factors | Child not deprived (%) | Child Deprived (%) | Chi-square | p-value |
|---|---|---|---|---|
| **Place of Residence** | | | | |
| Urban | 445 (5.0) | 8,469 (95.0) | 469.632 | 0.000 |
| Rural | 224 (1.0) | 21,666 (99.0) | | |
| **Other Factors** | | | | |
| **Exposure to Media** | | | | |
| No | 06 (0.2) | 3,472 (99.8) | 73.761 | 0.000 |
| Yes | 663 (2.4) | 26,663 (97.6) | | |
| **Orphanage Status** | | | | |
| No Parent Alive | 0 (0) | 94 (100) | 2.257 | 0.521 |
| Father Alive | 07 (2.5) | 277 (97.5) | | |
| Mother Alive | 13 (2.0) | 623 (98.0) | | |
| Both Parent Alive | 649 (2.2) | 9,138 (97.8) | | |
| **Relationship to Household Head** | | | | |
| Parent | 616 (2.3) | 26,604 (97.7) | 10.135 | 0.006 |
| Relatives | 50 (1.4) | 3,429 (98.6) | | |
| Others | 03 (2.9) | 102 (97.1) | | |

household head, the number of children, wealth index, place of residence and the relationship of the child to the household head were all significantly associated with the level of child health deprivation (p < 0.05).

## Multivariable analysis

Table 4 showed that in the North Central region, having the level of education of the household head to be primary level connotes 45% lower odds of the child experiencing child health deprivation as compared to those that have no education. Whereas household heads with secondary and tertiary have lower odds of children experiencing health deprivation as compared to those with no level of education (60% and 30% respectively). Household heads with the age range of 25–34 have about 29% higher odds of the child experiencing child health deprivation as compared to those that are in the age range of 15–24 years. While household heads with age range 35–44, 45–59 years, and 60 and above have lower odds (65%, 52%, and 89%) of the child experiencing health deprivation. Households with a family size of 10 and above have a 64% higher likelihood of the child experiencing child health deprivation as compared to those that are below that family size. Also, households having many children between 4–6 and 6 and above have more than 1 times and 3 times higher likelihood of the child experiencing health deprivation as compared to those that have between 0–3 children. Whereas in the context of the place of residence, respondents living in the rural area have 40% higher odds of children experiencing health deprivation as compared to those residing in the Urban areas. Similarly, being exposed to media has a higher likelihood of children experiencing health deprivation in the North Central region of Nigeria. Also, children who are relatives to the head of the household have a 66% higher likelihood of experiencing child health deprivation as compared to those who have a parental relationship with the household head. However, at the combined effect of all these factors, only two of them (household head having a tertiary level of education, Wealth index being at the richest class) were significantly associated with a child experiencing health deprivation in the North Central region (p < 0.05).

In the Northeast, only five factors were eminent at the bivariate level, having the level of education of the household head to be primary and secondary connotes 51% and 60% lower odds of the child experiencing health deprivation as compared to those that have none. Whereas respondents whose household heads had tertiary education had 87% lower odds of the child experiencing health deprivation as compared to those with no level of education. Households with a

**Table 3. Bivariate distribution of respondents by factors associated with child health deprivation in Nigeria.**

| Factors | Regions | | | | | |
|---|---|---|---|---|---|---|
| | North Central | Northeast | Northwest | Southeast | South-South | Southwest |
| **Childs Age** | | | | | | |
| 0 | 97.8 | 98.5 | 99.3 | 99.0 | 96.4 | 97.4 |
| 1 | 97.6 | 97.8 | 98.9 | 98.9 | 97.5 | 95.8 |
| 2 | 97.0 | 97.5 | 98.9 | 99.5 | 96.6 | 95.2 |
| 3 | 96.9 | 98.4 | 99.1 | 98.8 | 97.1 | 96.7 |
| 4 | 96.9 | 97.7 | 98.5 | 99.5 | 96.3 | 95.6 |
| $x^2$ | **3.8** | **6.1** | **5.8** | **2.8** | **2.2** | **5.7** |
| **Education of Household Head** | | | | | | |
| None | 99.5 | 99.6 | 99.8 | 99.3 | 98.6 | 99.3 |
| Primary | 98.8 | 99.0 | 99.8 | 99.9 | 97.7 | 99.5 |
| Secondary | 98.2 | 98.9 | 99.2 | 99.6 | 96.7 | 96.8 |
| Tertiary | 91.5 | 90.1 | 92.8 | 95.1 | 94.4 | 90.0 |
| $x^2$ | **217.8 \*** | **405.4\*** | **350.4 \*** | **64.6 \*** | **13.6 \*** | **104.8 \*** |
| **Sex of Household Head** | | | | | | |
| Male | 97.2 | 98.0 | 98.9 | 99.1 | 96.5 | 95.9 |
| Female | 97.1 | 95.5 | 99.2 | 99.4 | 97.6 | 98.0 |
| $x^2$ | **0.0** | **5.9 \*\*** | **0.1** | **0.5** | **1.8** | **4.4 \*** |
| **Age of Household Head** | | | | | | |
| 15–24 | 99.3 | 100.0 | 100.0 | 100.0 | 96.6 | 95.9 |
| 25–34 | 98.4 | 98.4 | 99.4 | 99.7 | 97.7 | 97.3 |
| 35–44 | 96.1 | 97.4 | 98.3 | 99.1 | 96.0 | 94.5 |
| 45–59 | 97.0 | 98.2 | 99.0 | 99.3 | 96.7 | 96.7 |
| 60 + | 98.8 | 98.0 | 99.6 | 98.6 | 97.5 | 98.8 |
| $x^2$ | **27.9 \*** | **8.1** | **18.7 \*** | **3.8** | **5.2** | **20.4 \*** |
| **Household Size** | | | | | | |
| 1–5 | 96.4 | 97.3 | 98.4 | 98.8 | 96.3 | 95.8 |
| 6–9 | 97.1 | 98.3 | 99.1 | 99.4 | 97.3 | 96.6 |
| 10+ | 99.1 | 98.0 | 99.2 | 100.0 | 96.9 | 96.8 |
| $x^2$ | **23.6 \*** | **6.3** | **7.7 (0.0)** | **3.9** | **2.3** | **1.6** |
| **Number of Children** | | | | | | |
| 0–3 | 96.6 | 97.9 | 98.8 | 99.0 | 96.7 | 95.9 |
| 4–6 | 98.6 | 98.0 | 99.1 | 100.0 | 96.9 | 98.9 |
| 7 and Above | 99.7 | 98.3 | 99.0 | 100.0 | 100.0 | 100.0 |
| $x^2$ | **21.9 \*** | **0.5** | **1.4** | **4.2** | **0.7** | **6.5 \*** |
| **Wealth index** | | | | | | |
| Poorest | 9.6 | 99.6 | 99.9 | 100.0 | 99.0 | 99.7 |
| Second | 99.6 | 99.7 | 99.7 | 100.0 | 99.0 | 99.7 |
| Middle | 98.5 | 98.0 | 99.4 | 100.0 | 98.4 | 98.7 |
| Fourth' | 97.6 | 93.3 | 97.8 | 100.0 | 95.2 | 97.5 |
| Richest | 83.7 | 82.1 | 86.3 | 96.6 | 91.8 | 91.1 |
| $x^2$ | **565.9 \*** | **535.5 \*** | **582.4 \*** | **47.5 \*** | **69.7 \*** | **115.9 \*** |
| **Place of Residence** | | | | | | |
| Urban | 92.6 | 96.5 | 96.9 | 96.1 | 91.4 | 95.1 |

*(Continued)*

**Table 3.** (Continued)

| Factors | Regions | | | | | |
|---|---|---|---|---|---|---|
| | **North Central** | **Northeast** | **Northwest** | **Southeast** | **South-South** | **Southwest** |
| Rural | 99.0 | 98.6 | 99.5 | 99.8 | 98.3 | 98.0 |
| **$\chi^2$** | **191.4 \*** | **34.7 \*** | **80.6 \*** | **63.1 \*** | **83.9 \*** | **17.6 \*** |
| **Exposure to Media** | | | | | | |
| No | 100.0 | 99.7 | 100.0 | 100.0 | 99.2 | 99.2 |
| Yes | 97.0 | 97.6 | 98.7 | 99.1 | 96.6 | 96.0 |
| **$\chi^2$** | **15.2 \*** | **20.3 \*** | **16.9 \*** | **1.8** | **5.0 \*** | **3.0** |
| **Relationship with Household Head** | | | | | | |
| Parent | 97.1 | 98.0 | 98.9 | 99.2 | 96.4 | 95.8 |
| Relatives | 98.6 | 97.9 | 99.5 | 98.9 | 98.3 | 98.6 |
| Others | 93.3 | 100.0 | 100.0 | 100.0 | 92.0 | 100.0 |
| **$\chi^2$** | **5.9 \*\*** | **0.4 \*** | **1.6** | **0.7** | **7.9 \*** | **8.4\*** |

*p<0.001; **p<0.05.

female as the head of the house have 77% lower odds of the child experiencing child health deprivation as compared to those that are male. Households with an average level of wealth index and more have higher odds (70%, 89%, and 95%) of the child not experiencing health deprivation as compared to those of the poorest households. Whereas respondents with households with a second-class wealth index have about 76% higher odds of the child experiencing health deprivation as compared to those that are of the poorest classes. Whereas in the context of place of residence, respondents living in the rural area have 54% higher odds of children experiencing health deprivation as compared to those residing in the Urban areas. Similarly, being exposed to media has a 41% lower likelihood of children experiencing health deprivation in the Northeast region of Nigeria. However, at the combined effect of all these factors, the household head having a tertiary level of education, female as the household head, Wealth index at middle, fourth, and richest, and living in a rural area were significantly associated with a child experiencing health deprivation in North East region (p<0.05).

Furthermore, the multivariable analysis revealed that in the North West, only five factors were eminent at the bivariate level in which, having the level of education of the household head to be primary is 1 times higher likelihood of the child experiencing health deprivation as compared to those that have none. Whereas household heads with secondary and tertiary have 45% and 85% lower odds of the child experiencing health deprivation as compared to those with no level of education. Households with a wealth index in the second class and above have lower odds of the child experiencing child health deprivation as compared to that of the poorest. Household heads with the age range of 25–34 and 60 and above have 1 time more likelihood of the child experiencing health deprivation as compared to those that are in the age range 15–24 years. While household heads with the age range 35–44 and 45–59 years have lower odds (52% and 43%) of the child experiencing health deprivation. The household size was also an important factor identified in this region as having a household size greater than 5 or more poses just 1 time more likelihood of child health deprivation as compared to those below 5. In the context of the place of residence respondents living in rural areas have a 7% higher likelihood of a child experiencing health deprivation as compared to those residing in the urban areas. However, at the combined effect of all these factors, only two variables, (household heads having a tertiary level of education, and wealth index at fourth and richest) were significantly associated with a child experiencing health deprivation in North West region (p<0.05).

In the South East, households having their number of children to be between 4–6 and 7 and above have 34% and 79% lower odds of the child experiencing health deprivation. Households with an average wealth index have a 2 times higher likelihood of the child experiencing health deprivation as compared to that of the poorest. Similarly, households with a wealth index of the fourth and richest have 99% and 40% higher odds of the child experiencing child health deprivation as

**Table 4. Multivariate analysis of the factors associated with child health deprivation in Nigeria.**

| Factors | Regions | | | | | | All Regions |
|---|---|---|---|---|---|---|---|
| | North Central OR (95% CI) | Northeast OR (95% CI) | Northwest OR (95% CI) | Southeast OR (95% CI) | South-South OR (95% CI) | Southwest OR (95% CI) | OR (95% CI) |
| **Childs Age** | | | | | | | |
| 0 | 1.0 (RC) | 1.0 (RC) | 1.0 (RC) | 1.0 (RC) | 1.0 (RC) | 1.0 (RC) | 1.0 (RC) |
| 1 | 1.14 (0.70-1.88) | 0.72 (0.39-1.32) | 1.34 (0.78-2.30) | 1.01 (0.59-1.76) | 0.85 (0.42-1.71) | 0.59 (0.36-0.99) | 0.81 (0.61-1.07) |
| 2 | 0.97 (0.60-1.56) | 0.86 (0.46-1.59) | 0.76 (0.47-1.22) | 0.76 (0.45-1.29) | 0.72 (0.37-1.39) | 0.89 (0.52-1.52) | 0.67 (0.52-0.88)* |
| 3 | 1.07 (0.67-1.72) | 0.64 (0.36-1.15) | 0.92 (0.57-1.49) | 0.92 (0.54-1.57) | 0.78 (0.40-1.51) | 0.67 (0.41-1.10) | 0.83 (0.63-1.09) |
| 4 | 0.91 (0.57-1.44) | 0.65 (0.37-1.17) | 0.89 (0.55-1.45) | 0.73 (0.44-1.21) | 0.53 (0.29-0.98) | 0.69 (0.42-1.12) | 0.65 (0.50-0.85)* |
| **Household Head Education** | | | | | | | |
| None | 1.0 (RC) | 1.0 (RC) | 1.0 (RC) | 1.0 (RC) | 1.0 (RC) | 1.0 (RC) | 1.0 (RC) |
| Primary | 0.55 (0.22-1.35) | 0.51 (0.22-1.17) | 1.57 (0.32-7.66) | 0.95 (0.56-1.60) | 0.69 (0.20-2.37) | 2.55 (0.50-13.09) | 0.59 (0.38-0.91)* |
| Secondary | 0.60 (0.27-1.35) | 0.60 (0.30-1.21) | 0.55 (0.20-1.53) | 1.08 (0.59-1.99) | 0.79 (0.23-2.67) | 0.67 (0.19-2.35) | 0.46 (0.31-0.68)* |
| Tertiary | 0.30 (0.14-0.67)* | 0.13 (0.07-0.23)* | 0.15 (0.06-0.40)* | 0.49 (0.24-0.98) | 0.77 (0.22-2.71) | 0.32 (0.09-1.14) | 0.19 (0.13-0.28)* |
| **Household Head Sex** | | | | | | | |
| Male | 1.0 (RC) | 1.0 (RC) | 1.0 (RC) | 1.0 (RC) | 1.0 (RC) | 1.0 (RC) | 1.0 (RC) |
| Female | 0.80 (0.49-1.32) | 0.23 (0.11-0.52)* | 1.33 (0.61-2.90) | 0.78 (0.53-1.17) | 0.93 (0.58-1.48) | 1.06 (0.49-2.32) | 0.77 (0.63-0.93) |
| **Household Head Age** | | | | | | | |
| 15–24 | 1.0 (RC) | 1.0 (RC) | 1.0 (RC) | 1.0 (RC) | 1.0 (RC) | 1.0 (RC) | 1.0 (RC) |
| 25–34 | 1.29 (0.17-9.89) | 2.62 (0.76-9.04) | 1.17 (0.31-4.46) | 0.91 (0.43-1.92) | 0.99 (0.45-2.22) | 3.21 (0.88-11.66) | 1.68 (0.72-3.91) |
| 35–44 | 0.65 (0.09-4.85) | 2.77 (0.82-9.36) | 0.48 (0.16-1.46) | 0.98 (0.58-1.65) | 0.90 (0.49-1.63) | 1.90 (0.54-6.65) | 1.09 (0.47-2.50) |
| 45–59 | 0.52 (0.07-3.96) | 2.88 (0.85-9.77) | 0.57 (0.19-1.69) | 1.07 (0.68-1.68) | 1.16 (0.65-2.06) | 2.06 (0.55-7.63) | 1.04 (0.44-2.42) |
| 60 + | 0.89 (0.10-7.91) | 3.31 (0.97-11.35) | 1 | 1 | 1 | 3.55 (0.65-19.31) | 1.07 (0.43-3.62) |
| **Household Size** | | | | | | | |
| 1–5 | 1.0 (RC) | 1.0 (RC) | 1.0 (RC) | 1.0 (RC) | 1.0 (RC) | 1.0 (RC) | 1.0 (RC) |
| 6–9 | 0.95 (0.66-1.36) | 1.05 (0.65-1.68) | 1.43 (0.77-2.65) | 0.90 (0.61-1.33) | 1.03 (0.66-1.60) | 1.13 (0.77-1.66) | 1.13 (0.93-1.37) |
| 10+ | 1.64 (0.72-3.71) | 2.55 (1.17-5.53) | 1.62 (0.79-3.31) | 0.85 (0.34-2.15) | 2.24 (0.50-10.06) | 0.73 (0.23-2.28) | 1.09 (0.75-1.58) |
| **Number of Children** | | | | | | | |
| 0-3 | 1.0 (RC) | 1.0 (RC) | 1.0 (RC) | 1.0 (RC) | 1.0 (RC) | 1.0 (RC) | 1.0 (RC) |
| 4-6 | 1.56 (0.87-2.82) | 0.68 (0.42-1.09) | 0.81 (0.54-1.22) | 0.66 (0.45-0.95)* | 0.66 (0.38-1.14) | 1.44 (0.43-4.76) | 1.19 (0.88-1.59) |
| 7 and Above | 3.43 (0.41-28.59) | 0.39 (0.16-0.95) | 1.49 (0.68-3.28) | 0.21 (0.04-1.02) | 1 | 1 | 1.50 (0.88-2.54) |
| **Wealth index** | | | | | | | |
| Poorest | 1.0 (RC) | 1.0 (RC) | 1.0 (RC) | 1.0 (RC) | 1.0 (RC) | 1.0 (RC) | 1.0 (RC) |
| Second | 1.23 (0.41-3.69) | 1.76 (0.64-4.88) | 0.39 (0.08-1.97) | 0.91 (0.56-1.48) | 1.26 (0.31-5.14) | 1.22 (0.07-20.02) | 1.00 (0.57-1.77) |

*(Continued)*

**Table 4.** (Continued)

| Factors | Regions | | | | | | All Regions OR (95% CI) |
|---|---|---|---|---|---|---|---|
| | North Central OR (95% CI) | Northeast OR (95% CI) | Northwest OR (95% CI) | Southeast OR (95% CI) | South-South OR (95% CI) | Southwest OR (95% CI) | |
| Middle | 0.46 (0.18-1.16) | 0.30 (0.14-0.64)* | 0.26 (0.05-1.32) | 2.05 (1.18-2.57)* | 0.91 (0.24-3.43) | 0.21 (0.02-1.77) | 0.36 (0.22-0.59)* |
| Fourth' | 0.41 (0.16-1.07) | 0.11 (0.05-0.23)* | 0.14 (0.03-0.72)* | 1.99 (1.12-3.57)* | 0.41 (0.11-1.54) | 0.13 (0.02-1.06) | 0.17 (0.10-0.28)* |
| Richest | 0.07 (0.03-0.19)* | 0.05 (0.02-0.11)* | 0.03 (0.00-0.17)* | 1.40 (0.73-2.68) | 0.32 (0.08-1.21) | 0.05 (0.01-0.37)* | 0.06 (0.04-0.10)* |
| **Place of Residence** | | | | | | | |
| Urban | **1.0 (RC)** | **1.0 (RC)** | **1.0 (RC)** | **1.0 (RC)** | **1.0 (RC)** | **1.0 (RC)** | **1.0 (RC)** |
| Rural | 1.40 (0.91-2.17) | 0.54 (0.37-0.78)* | 1.07 (0.64-1.80) | 7.17 (2.17-23.74)* | 3.14 (1.99-4.96)* | 0.70 (0.42-1.17) | 1.15(0.95-1.40) |
| **Exposure to Media** | | | | | | | |
| No | **1.0 (RC)** | **1.0 (RC)** | **1.0 (RC)** | **1.0 (RC)** | **1.0 (RC)** | **1.0 (RC)** | **1.0 (RC)** |
| Yes | 1.34 (0.49-3.68) | 0.59 (0.17-2.05) | 1.43 (0.87-2.34) | 1.36 (0.79-2.33) | 0.58 (0.12-2.70) | 1.39 (0.55-3.49) | 0.49(0.21-1.15) |
| **Relationship with Household Head** | | | | | | | |
| Parent | **1.0 (RC)** | **1.0 (RC)** | **1.0 (RC)** | **1.0 (RC)** | **1.0 (RC)** | **1.0 (RC)** | **1.0 (RC)** |
| Relatives | 1.66 (0.76-3.62) | 1.10 (0.63-1.92) | 1.27 (0.68-2.38) | 1.53 (0.95-2.49) | 1.78 (0.93-3.42) | 1.48 (0.55-3.97) | 1.20(0.85-1.69) |
| Others | 0.48 (0.05-4.35) | 1 | 0.49 (0.14-1.66) | 1.69 (0.64-4.52) | 0.41 (0.09-1.90) | 1 | 0.74(0.22-2.49) |

**\*** p<0.05

compared to that of the poorest. Children living in rural areas are 7 times more likely to experience child health deprivation as compared to those residing in urban areas. However, at the combined effect of all these factors, having between 4–6 numbers of children and having a wealth index at average and fourth class and place of residence were significantly associated with a child experiencing health deprivation in the Southeast region (p<0.05).

Also, results from the South-South region revealed that only five factors were eminent at the bivariate level in which, having the level of education of the household head to be primary, secondary, and tertiary connotes 31%, 21%, and 23% lower odds of the child experiencing health deprivation as compared to those that have no level of education. Households with wealth index that are second class have a 1 time more likelihood of the child experiencing health deprivation as compared to those of the poorest households. Whereas children living in rural areas are 3 times more likely to experience poverty as compared to those living in the urban area. Similarly, being exposed to media has a 58% higher likelihood of children experiencing health deprivation. Also, children who have their relationship with the household head to be relatives have a 78% higher likelihood of experiencing child health deprivation as compared to those who have a parental relationship with the household head. However, at the combined effect of all these factors, the household living in the rural area was the only significant factor associated with a child experiencing health deprivation in the South-South region (p<0.05).

Furthermore, results from the multivariable analysis for the regions also revealed that in South West only seven factors were eminent at the bivariate level which formed the basis for the analysis, having the level of education of the household head be primary connotes 2 times more likelihood of a child been health deprived as compared to those with no education. Whereas having a secondary and tertiary level of education connotes lower odds (33% and 68%) of the child experiencing health deprivation as compared to those that have no education. Household heads being females have a 6% higher likelihood of child health deprivation as compared to those that have male household heads. Household

heads with an age range of 25–34 and 60 and above years have 3 times higher odds of the child experiencing health deprivation as compared to those that are in the age range of 15–24 years. Household heads with the age range 35–44 and 45–59 years have a 2 times higher likelihood of the child experiencing health deprivation as compared to those with household head age 15–24. Households having many children to be between 4–6 and 7 and above have 1 times higher odds of the child experiencing health deprivation. Households with a wealth index second have higher odds (22%) of the child not experiencing health deprivation as compared to those that are in the poorest class. Whereas households with a wealth index of average and greater have higher likelihoods (79%, 87%, and 95%) of the child not experiencing health deprivation as compared to those in the poorest class. In the context of place of residence children living in the rural area have a 70% higher likelihood of experiencing health deprivation as compared to those residing in the urban areas. Also, children who have their relationship to the household head to be relatives or others are 1 time more likely to experience child health deprivation as compared to those that have a parental relationship with the household head. However, at the combined effect of all these factors, only the wealth index of the household been at the richest class was significantly associated with a child experiencing health deprivation in South West region ($p < 0.05$). At the general basis comprising all the regions, the multivariable analysis result explains the combined relationship between the independent variables and the dependent variable identified in the study. It could be observed that children who are 1 year of age have an 81% higher likelihood of child health deprivation as compared to those who are age 0. Similarly, children who are above 1 year of age have a higher likelihood (67%, 83%, and 65%) of child health deprivation as compared to those who are age 0. Also, having the level of education of the household head to be primary, secondary, and tertiary connotes a 41%, 54%, and 81% higher odds of the child not experiencing health deprivation as compared to those that have none. Household heads with an age range of 25–34 have higher odds (68%) of the child experiencing health deprivation as compared to those that are in the age range of 15–24 years. While household heads with age range 35–44, 45–59, and 60 and above years have higher odds (9%, 4%, and 7%) of the child experiencing health deprivation. Households with family size 6–9 and 10 and above have a 13% and 9% higher likelihood of the child experiencing child health deprivation as compared to those that have family size 1–5. Also, households having many children to be between 4–6 and 7 and above have a 19% and 50% higher likelihood of the child experiencing health deprivation. While households with a wealth index at second class are 1 times higher likelihood, those that are greater than the second class have higher likelihoods (64%, 83%, 64%, and 94%) of the child not experiencing health deprivation as compared to that of the poorest. Also, in the context of place of residence respondents living in rural areas have 15% higher odds of experiencing health deprivation as compared to those residing in urban areas. Similarly, being exposed to media connotes a 51% lower likelihood of a child not experiencing health deprivation. Also, in the context of children's relationship with the household head, those who are relatives have a 20% higher likelihood of experiencing health deprivation as compared to those who are related to the household head as parents. Also. While still examining the children's relationship with household head factor, those that are of other relationship have 26% lower likelihoods of child not experiencing health deprivation as compared to those that are related to household head as parents. However, at the combined effect of all these factors, only four out of ten of the variables (child age at 2 and 4 years, household head level of education, Wealth index greater than the average class, and geopolitical zone) were significantly associated with a child experiencing health deprivation in all region ($p < 0.05$).

## Discussion of findings

The study examined the prevalence of child health deprivation and its associated factors among under-five children at the regional and national levels in Nigeria. The study examined the prevalence of child health deprivation across regions separately and jointly. Results show that the level of child health deprivation nationally is high, and this is reflected when broken into regions with slight variations across regions. Nationally, evidence shows that nearly all children (97%) experience at least a deprivation in healthcare. This figure is significantly higher than that in developed nations, where child health deprivation is around 10% [22]. Compared to other African regions, the figure in this study is higher compared to

other African countries such as Tanzania and Kenya [26,27]. A study conducted in Nigeria using Nigeria Demographic Data for 2008, reported a child health deprivation prevalence rate of 79.2% [28]. This disparity may be attributed to limited health interventions addressing key issues such as access to healthcare, access to full immunization against diseases, poverty, and insurgencies. Previous studies support this view, highlighting that pervasive poverty and inadequate healthcare infrastructure contribute to delays in healthcare uptake by parents, thereby exacerbating Child health deprivation [29,30]. Similarly, a study [31] indicated that children born into poorer families have lower survival rates compared to those from wealthier families. This disparity is attributed to various factors, including inadequate nutrition and limited access to healthcare services.

Interestingly, the regional prevalence rates align closely with the national average, except for the Northern regions, which are slightly higher. This finding has been explained by the CSCFH. The Northern regions is more conservative. The region is characterized by a strong adherence to promote traditional institutions, customs and values. The Northern region is resistance to change in the socio-cultural, political philosophy and ideology. Thus, the region may face little progress or failure in healthcare services, particularly when it comes to adopting new ideas or innovations that are perceived to contradicting some aspects of the traditional culture. Other factors contributing to this discrepancy may include the increased insurgency in the area, which complicates the utilization of healthcare insurance schemes. Residents often consider the insecurity and the distance to healthcare facilities, leading them to rely on local or substandard services that are more accessible for child health. Additionally, the high poverty rate and rural residence of many individuals in the Northern region may affect their willingness to register for and utilize healthcare insurance schemes for their children and as well receive full doses of the immunization packages upon child delivery. A situation better explained using CSCFH. In corroboration to this perspective, [32] noted that the Northern region experienced a rise in child abductions in 2018, with children being recruited by armed groups. Such incidents may make parents hesitant to leave their homes, contributing to child health deprivation in the region. The above stated factors lent credence to the Cosmopolitan-Success and Conservative-Failure Hypothesis (CSCFH) that explain gap between the North and the South in their willingness to register and utilize healthcare insurance scheme for their children.

Despite the availability of health insurance schemes at both federal and state levels intended to reduce costs for parents, especially the poor, evidence suggests low enrollment due to organizational challenges [33,34]. Although our results indicate high media exposure among household heads, the majority of whom are underprivileged (95%), awareness of these schemes may be low due to insufficient governmental promotion. Furthermore, stringent enrollment criteria may limit uptake, particularly in rural areas where over half of the children reside. A study [35] concluded that, despite the intervention's potential benefits, the coverage of enrollees' families was relatively poor, thereby hindering the scheme's effectiveness. Furthermore, despite efforts by the government to increase campaigns promoting child immunization and proper care during fever, the achieved figures remain high and do not reflect significant improvement. The CSCFH explained the reason for relatively poor enrolment and insignificant improvement in the North despite government efforts. It is expected that conservative societies characterized by a strong adherence to promote traditional institutions, customs and values, and resistance to change in the socio-cultural, political philosophy and ideology) may face sluggish or little progress or failure, particularly when it comes to adopting new ideas or innovations that are perceived to contradicting some aspects of the traditional culture. In correlation to this thought [36] identified significant gaps in the evidence regarding the delivery of effective and sustainable routine childhood immunization. A recent analysis of urban poor and child health deprivation in Nigeria mentioned that the prevalence of zero-dose children (those aged 12–23 months who have not received the DPT 1 vaccine) is higher among urban poor children (35%) compared to urban non-poor children (15%), with the highest prevalence observed among rural children (45%) [37]. This phenomenon gives credence to the CSCFH that explained differences between the North and the South in the uptake of DPT 1 vaccine.

The limitations of using secondary data may have influenced the results, as the actual number of enrollees and service users might not have been accurately captured. However, previous studies have highlighted the substantial benefits

of health insurance utilization, ensuring quality healthcare for both children and adults [38]. In contrast, another study [20] pointed out that cultural and traditional issues can hinder parents' adherence to medical care for their children. This situation is tandem with the CSCFH as negative attitudes towards Western medical care may be more prevalent in the North than in the South. Analyzing factors associated with under-five child health deprivation across regions revealed that the wealth index, level of education of the household head, and place of residence significantly influenced under-five child health deprivation in Nigeria (p<0.05). This pattern persisted across regions, except in the Northeast where the household head sex, and Southeast, where the number of children in the household also emerged as a significant factor. This discrepancy may be due to societal perceptions of these regions, influencing the level of care provided. For example, in the Southeast which they have high tendencies of increased number of children who household mostly poor, under-five children in such households are more likely to receive less health care services in the hospital, as such opt for self-care at home, a practice not as prevalent in other regions. This situation is in tandem with CSCFH that explain the negative attitude of the Northerners towards western healthcare services in the hospital and as such opt for self-care at home compared with the Southerners in Nigeria.

The education level of the household head plays a crucial role in determining child health outcomes. In Nigeria, the household head is typically male and the primary decision-maker, meaning their level of education directly impacts the well-being of children in the household. Higher education levels in household heads correlate with lower instances of under-five child health deprivation both regionally and nationally. This underscores the importance of the household head's understanding of quality-of-life dimensions. Contrary to this, a study in Turkey found a negative association between the household head's education and child health deprivation [39]. Additionally, the age of the household head influences under-five child health deprivation, as older heads may delegate certain responsibilities to children, potentially increasing their deprivation, especially in the South West region which results revealed a higher likelihood for under-five children experiencing health deprivation. For instance, older heads might assign tasks such as fetching water to younger children, reflecting a view of children as contributors to household needs rather than recipients of care, thus increasing deprivation during their growth [20].

Also, in ascertaining the association between household factors and child health deprivation both across all regions and individually, results revealed that household size, number of living children, wealth index, zone, and place of residence were key contributors to child health deprivation across all regions. However, when analyzed regionally, the wealth index and place of residence were the most consistent factors significant in all the regions, except in North Central (household size, number of children) and South East (number of children), with the number of children in the South East being significant at the multivariate level. In the South East, a higher number of children increases the likelihood of health deprivation due to the strain on family resources. In North Central, children in families with more than 10 members have a 64% higher chance of experiencing health deprivation, likely due to divided attention and resources. This phenomenon supports the assumption of CSCFH and is less likely to occur in the South. Previous studies [40,41] have also found that large family sizes can lead to increased stress and reduced well-being.

The household wealth index is another significant factor, with poorer households having a higher likelihood of child health deprivation. Interestingly, in the Southeast, even households with average or above-average wealth had higher odds of child health deprivation compared to the poorest households, possibly due to differences in health insurance coverage or level of advice south when they are under five children experience fever or diarrhoea. Generally, economic downturns can cause households to move from higher to lower wealth levels, affecting their access to essential social amenities, particularly in rural areas. Previous studies [42,43] affirmed that rural areas often lack quality social amenities crucial for child development. A study [44] found a higher likelihood of child survival linked to household wealth status. Previous studies [29,41,44–47] have also established that children in rural areas have a higher tendency for deprivation due to limited resources. However, it should be noted that regional differences in child health deprivation are complex and are influenced by various factors beyond just cosmopolitanism and conservatism.

Furthermore, factors such as media exposure and relationship with the household head were also significant at the bivariate level, with children raised by relatives having a higher likelihood of experiencing health deprivation as compared to those raised by their parents. This highlights the serious health deprivation for under-five children in Nigeria and occasions of not being addressed increase the possibility of having an increased under-five mortality cases in Nigeria.

## Strengths and limitations

The study gives insight into the dimensions of child health deprivation from the perspective of secondary data in Nigeria. MICS is a national data which makes it easy to generalize the findings to all geopolitical zones. However, the survey was retrospective, therefore memory lapses can occur. Also, MICS is a cross-sectional survey thus, it is difficult to establish a cause-effect relationship among the variables examined. Finally, the use of a secondary data source might have limits on the number of factors that can be examined in having a full glimpse of the factors associated with child health deprivation.

## Conclusion

Child health deprivation among children is widespread across all regions of Nigeria, primarily influenced by household factors as compared to individual factors. In all regions, the education of household head, wealth index and place of residence influence deprivation in health for children, with the influence of other factors varying by region. This accentuates the need for a comprehensive review of policies and strategies related to health insurance schemes, vaccination programs, and treatment services for diarrhoea and fever, particularly targeting under-five children. While the study identifies significant contributing factors, it notes the potential variability in findings from studies using primary data sources. Furthermore, the analysis may have been limited by the range of factors available from the current data source, indicating that future research should consider a broader array of household determinants impacting child health deprivation.

## Acknowledgments

The authors are grateful to the National Bureau of Statistics (NBS) [Nigeria] and United Nations Children's Fund (UNICEF) for the permission to use the datasets. The views expressed in the publication represent those of the authors and do not necessarily represent the official views of the National Bureau of Statistics (NBS) [Nigeria] and United Nations Children's Fund (UNICEF).

## Author contributions

**Conceptualization:** Victor Chima, Funmilola F. Oyinlola, Joseph Ayodeji Kupoluyi.

**Data curation:** Victor Chima, Segun Tekun.

**Formal analysis:** Victor Chima, Funmilola F. Oyinlola, Joseph Ayodeji Kupoluyi, Segun Tekun, Ifeyinwa U. Anyanyo.

**Investigation:** Victor Chima.

**Methodology:** Victor Chima, Funmilola F. Oyinlola, Joseph Ayodeji Kupoluyi, Segun Tekun, Ifeyinwa U. Anyanyo.

**Supervision:** Victor Chima, Funmilola F. Oyinlola, Joseph Ayodeji Kupoluyi.

**Writing – original draft:** Victor Chima, Funmilola F. Oyinlola, Joseph Ayodeji Kupoluyi, Segun Tekun, Ifeyinwa U. Anyanyo.

**Writing – review & editing:** Victor Chima, Funmilola F. Oyinlola, Joseph Ayodeji Kupoluyi, Segun Tekun, Ifeyinwa U. Anyanyo.

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
