## [Decision Letter · Decision Letter 0]

PGPH-D-24-01755

Regional variations in child health deprivation and its associated factors in Nigeria.

Dear Dr. Kupoluyi,

Thank you for submitting your manuscript to PLOS Global Public Health. After careful consideration, we feel that it has merit but does not fully meet PLOS Global Public Health’s publication criteria as it currently stands. Therefore, we invite you to submit a revised version of the manuscript that addresses the points raised during the review process.

We look forward to receiving your revised manuscript.

Kind regards,

Opeyemi Oluwatosin Babajide, Ph.D.

Academic Editor

Journal Requirements:

1. Please note that your Data Availability Statement is currently missing [the repository name and/or the DOI/accession number of each dataset OR a direct link to access each database]. If your manuscript is accepted for publication, you will be asked to provide these details on a very short timeline. We therefore suggest that you provide this information now, though we will not hold up the peer review process if you are unable.

2. Please provide separate figure files in .tif or .eps format.

3. Tables should not be uploaded as individual files. Please remove these files and include the Tables in your manuscript file as editable, cell-based objects. For more information about how to format tables, see our guidelines:

https://journals.plos.org/globalpublichealth/s/tables

Additional Editor Comments (if provided):

Reviewers' comments:

Reviewer's Responses to Questions

**Comments to the Author**

1. Does this manuscript meet PLOS Global Public Health’s publication criteria ? Is the manuscript technically sound, and do the data support the conclusions? The manuscript must describe methodologically and ethically rigorous research with conclusions that are appropriately drawn based on the data presented.

Reviewer #1: Yes

Reviewer #2: Yes

2. Has the statistical analysis been performed appropriately and rigorously?

Reviewer #1: Yes

Reviewer #2: No

3. Have the authors made all data underlying the findings in their manuscript fully available (please refer to the Data Availability Statement at the start of the manuscript PDF file)?

Reviewer #1: Yes

Reviewer #2: Yes

4. Is the manuscript presented in an intelligible fashion and written in standard English?

Reviewer #1: Yes

Reviewer #2: Yes

5. Review Comments to the Author

Reviewer #1: The introduction is lengthy.

The research objectives are mentioned only briefly at the end. It would be more effective to explicitly state the aims earlier on and reinforce them throughout. Specifying that the study will address regional variations in child health deprivation would give the reader a more immediate understanding of the study’s focus. The introduction structure feels a bit fragmented. Reorganizing the content into well-defined sections—global context, Nigerian context, healthcare, regional disparities—would improve readability.

The introduction mentions that 333 million children are living in extreme poverty in 2022, but then states that the number had declined from 385 million, also in 2022. This seems contradictory and may confuse the reader. The timeline needs clarification (perhaps comparing different years more distinctly).

The data on child health deprivation in Nigeria could be structured more clearly. Several figures are presented but not always synthesized into a coherent narrative. It would help to summarize key points rather than listing multiple statistics consecutively, which could overwhelm the reader.

There is some repetition, particularly around the definitions of child deprivation. For example, both UNICEF’s and Child Fund International’s definitions are quoted, but they overlap significantly. The introduction could benefit from synthesizing these definitions into a more concise explanation.

The introduction does not identify the gap in the literature until the very end. While the last few sentences touch on this ("Studies on regional variation of child health deprivation... remain scarce"), it would be more impactful if this gap was highlighted earlier. The reader should know from the beginning what unique contribution this study intends to make.

The introduction jumps between global and national (Nigeria) data points without smooth transitions. For instance, the discussion on global child poverty transitions abruptly to Nigeria-specific data. A more logical progression would start broadly (global context), then gradually narrow down to Nigeria, offering transitional sentences to guide the reader.

Although the introduction is data-rich, it lacks adequate explanation or critical engagement with the numbers. For example, what are the reasons behind the regional variations in child health deprivation within Nigeria? Why do certain regions perform worse than others in terms of immunizations or health insurance? Some reflection on these trends would strengthen the narrative.

Method

While the selected health indicators are justified to some extent with references, the choice of the four specific indicators (penta 3, diarrhea, fever, and insurance coverage) could be further justified in the context of the broader health and deprivation literature. Why were these indicators chosen over others (e.g., malnutrition or stunting)? What is their relevance to child health deprivation in Nigeria specifically?

The inclusion of explanatory variables such as child age, sex, household head’s age, etc., is appropriate, but there’s no explanation of how these variables have been shown to affect health outcomes in previous research. Providing such a rationale would strengthen the justification for their inclusion.

The geopolitical zones are identified as the main explanatory variable, but there’s no discussion about why these zones are significant in relation to child health deprivation. Including a brief explanation of the socioeconomic, cultural, or health system differences across these regions would provide a stronger foundation for their use in the analysis.

While the methodology appropriately mentions incorporating design effects and survey weights, the description of how variances will be estimated using the Taylor Linearized variance estimation approach could benefit from further simplification. Non-technical readers may find this section difficult to follow. A brief explanation of why this method is chosen and how it affects the precision of estimates would improve clarity.

Discussion

While the study highlights higher deprivation rates in Northern regions due to insurgency, it does not provide an in-depth exploration of other regions' unique challenges. For example, the higher deprivation rate in the Southeast is attributed to household size, but other cultural or systemic factors that could play a role are not fully explored.

The study compares its findings to other African countries and previous research in Nigeria, but it could further explore why such disparities exist. For example, why is the child health deprivation rate in Nigeria much higher than in Kenya or Tanzania? Examining differences in health policies, economic structures, or international aid efforts would have added value to these comparisons.

Reviewer #2: I appreciate the opportunity to evaluate this manuscript. These are significant concerns, and I think that fixing them will improve the article's quality and acceptability for publishing. They are listed in the following order:

The authors should include a brief explanation of the specific child health indicators used in the, such as child’s age, sex, education of the household head, and household size, to provide more context for the study's focus, while also highlighting the regional variations examined. Listing the various descriptive statistical tools used in the analysis is irrelevant and can be simplified to state that "the data were analyzed using different descriptive statistics to examine regional variations in child health." Furthermore, since the study relies on secondary data, explaining the study design is unnecessary in the abstract. Instead, the authors should summarize the relevance of the results using statistical tests such as Pearson Chi-square and Binary Logistic Regression to strengthen the methodological accuracy. Additionally, the results section of the abstract need improvement, as it does not adequately report the findings, and the percentage prevalence of child health deprivation mentioned is not found in the methodology. Lastly, the authors should discuss potential policy recommendations or future research directions based on the results, and since diarrhea and fever were not included as variables in the analysis, they should not be mentioned in the conclusion regarding treatment services.

6. PLOS authors have the option to publish the peer review history of their article (what does this mean? ). If published, this will include your full peer review and any attached files.

**Do you want your identity to be public for this peer review?** For information about this choice, including consent withdrawal, please see our Privacy Policy .

Reviewer #1: **Yes: ** KHADIJAT ADELEYE

Reviewer #2: No

---

## [Decision Letter · Decision Letter 1]

PGPH-D-24-01755R1

Regional variations in child health deprivation and its associated factors in Nigeria.

Dear Dr. Kupoluyi,

Thank you for submitting your manuscript to PLOS Global Public Health. After careful consideration, we feel that it has merit but does not fully meet PLOS Global Public Health’s publication criteria as it currently stands. Therefore, we invite you to submit a revised version of the manuscript that addresses the points raised during the review process.

We look forward to receiving your revised manuscript.

Kind regards,

Miquel Vall-llosera Camps, Ph.D.

Staff Editor

Additional Editor Comments:

We note that one or more reviewers has recommended that you cite specific previously published works. As always, we recommend that you please review and evaluate the requested works to determine whether they are relevant and should be cited. It is not a requirement to cite these works. We appreciate your attention to this request.

Reviewers' comments:

Reviewer's Responses to Questions

**Comments to the Author**

1. If the authors have adequately addressed your comments raised in a previous round of review and you feel that this manuscript is now acceptable for publication, you may indicate that here to bypass the “Comments to the Author” section, enter your conflict of interest statement in the “Confidential to Editor” section, and submit your "Accept" recommendation.

Reviewer #3: (No Response)

2. Does this manuscript meet PLOS Global Public Health’s publication criteria ? Is the manuscript technically sound, and do the data support the conclusions? The manuscript must describe methodologically and ethically rigorous research with conclusions that are appropriately drawn based on the data presented.

Reviewer #3: Partly

3. Has the statistical analysis been performed appropriately and rigorously?

Reviewer #3: No

4. Have the authors made all data underlying the findings in their manuscript fully available (please refer to the Data Availability Statement at the start of the manuscript PDF file)?

Reviewer #3: Yes

5. Is the manuscript presented in an intelligible fashion and written in standard English?

Reviewer #3: Yes

6. Review Comments to the Author

Reviewer #3: The study investigates factors related to child health deprivation and how they vary across regions in Nigeria. Regions are defined in the study by geo-political zones. The study utilized the 2021 version of the Multiple Indicator Survey in Nigeria and found that basic and household characteristics were significant predictors of child health deprivation.

In all, the study is interesting, well-written and worthy of publication. However, there are fundamental issues that need to be addressed. I recommend a major revision of the manuscript before it is published by the Plos Global Public Health.

A fundamental issue with the manuscript is the lack of theoretical orientation or framework, and this is common among health scientists. The problem of regional variation, which the paper investigated, is a culturally and socially important one in Nigeria. It is a significant contribution that the study ought to make to scholarship. However, from the abstract to the conclusion, the issue was only discussed in statistical terms. Evidence in other studies shows that the north and south (and the geopolitical zones by extension) have significant socio-cultural differences, ultimately affecting health, economy and social activities. The study failed to adequately demonstrate why it is important to investigate regional differences in Nigeria. The academic world needs to know this! It is an important contribution to scholarship. I provide specific comments on each section of the manuscript

Abstract:

the introductory section of the abstract is a testament to the overall weakness of the manuscript. It says, “Nigeria has one of the worst child welfare metrics in the world, with child deprivation prevalent across all the social sectors. Children experience deprivation and poverty differently when compared to adults, and this ultimately affects their development. This study examined regional variations in child health deprivation and its associated factors in Nigeria.” Meanwhile, the issue of regional variations and its importance in Nigeria was not emphasized in the introduction but only in the results.

Background:

the background provides adequate information about the different forms of child deprivation and emphasizes child health deprivation in particular. However, the section did not do justice to the existence and importance of investigating regional differences in Nigeria. The mention of regional disparities in health and rural residence is not enough. The study claimed that “These studies somewhat focused on single indicators to measure child health deprivations or limited in its approach to examine regional variations by associated factors (p. 7)” but it also failed to explain why the academic world should care about regional variations in Nigeria. I have conducted research and collected primary data across geopolitical zones in Nigeria, and I encourage the authors to emphasize the socio-cultural differences across regions and geopolitical zones in the country, and how they affect health outcomes and consequently child health deprivation.

Since the introduction already appears too long, I encourage the authors to create a section for theoretical framework or “Importance of investigating regional variations in Nigeria.” My colleague in Nigeria has done some work in regional variations in Nigeria. The authors may utilize some of them if they find them necessary and relevant to their work.

https://doi.org/10.1017/S0021932023000238

https://doi.org/10.1017/S0021932022000463

https://doi.org/10.1017/S0021932020000747

That a manuscript is empirical does not mean it should be atheoretical. The authors may need to add that their study contributes to Sustainable Development Goal 10- Reduced inequalities.

Methods

The authors need to re-write how they operationalized the dependent variable- child health deprivation. They should tell the readers about the scoring system. It is not enough to say the composite measures were converted to binary outcomes. How were the four variables scored? Is it ‘0’ and ‘1’ for each of them? If yes, what score range was used to determine “child-deprived” and “child-not deprived”? The authors wrote that “The composite variable was computed into a binary dimension of child-deprived (those who were not covered by full immunization, did not receive diarrhea treatment, fever treatment, and were not covered by the health insurance scheme) and child not deprived (those who were covered by immunization, have received diarrhea treatment from the health facility, received fever treatment, and covered by the health insurance scheme)”. There were four indicators:

• full immunization

• diarrhea treatment

• Fever treatment

• Health insurance

How did you categorise those who had two or three out of these four? If my child had full immunization, received diarrhea treatment and fever treatment but did not have health insurance, would you say (s)he is deprived?

The authors wrote “The geopolitical zones were the main explanatory variable, and it was measured as zones.” Again, this explains why regional differences need to be grounded theoretically in the work. The authors need to link the relevance of “other explanatory variables” and “intervening variables” to the study. In large datasets, associational tests are likely to be statistically significant. The authors need to state how other variables logically relate to the main (in)dependent variable. The section on “statistical analyses” can be discussed at descriptive and inferential for the sake of coherence.

I respect your choice of statistics and analyses, but have you ever come across situations where variables are insignificant at a bivariate level but appear statistically significant in multivariate analyses?

Results:

the authors should show the mean and SD of the ratio variables (age, household size, number of children, etc). The label of the figures should be at the bottom, not inside the figures. Interestingly, the question I asked in the method popped up again in Table 1. Health insurance issues are complex in Nigeria; again, if my child had “yes” in the three other variables but “no” for health insurance, does that mean (s)he is deprived, considering the structure and nature of health insurance in Nigeria? The context of the location of your research matters!

SINCE 97% OF THE CHILDREN ARE DEPRIVED, ACCORDING TO THE TREATMENT OF YOUR COMPOSITE MEASURES, THEN WHAT IS THE BASIS FOR ASSOCIATIONAL ANALYSIS?

7. PLOS authors have the option to publish the peer review history of their article (what does this mean? ). If published, this will include your full peer review and any attached files.

**Do you want your identity to be public for this peer review?** For information about this choice, including consent withdrawal, please see our Privacy Policy .

Reviewer #3: No

---

## [Decision Letter · Decision Letter 2]

PGPH-D-24-01755R2

Regional variations in child health deprivation and its associated factors in Nigeria.

Dear Dr. Kupoluyi,

Thank you for submitting your manuscript to PLOS Global Public Health. After careful consideration, we feel that it has merit but does not fully meet PLOS Global Public Health’s publication criteria as it currently stands. Therefore, we invite you to submit a revised version of the manuscript that addresses the points raised during the review process.

Some further minor revisions have been requested: Reviewer 3 requested further discussion of the extent to which the assumptions of the CSCFH support/contradict the results of your study needs to be discussed in your results and discussion of findings. Which region has made progress and which has failed as far as child deprivation is concerned? To what extent is this a function of the region's socio-cultural and economic outlook and beliefs?

We look forward to receiving your revised manuscript.

Kind regards,

Jennifer Tucker, PhD

Staff Editor

Journal Requirements:

Additional Editor Comments (if provided):

Reviewers' comments:

Reviewer's Responses to Questions

**Comments to the Author**

1. If the authors have adequately addressed your comments raised in a previous round of review and you feel that this manuscript is now acceptable for publication, you may indicate that here to bypass the “Comments to the Author” section, enter your conflict of interest statement in the “Confidential to Editor” section, and submit your "Accept" recommendation.

Reviewer #3: All comments have been addressed

2. Does this manuscript meet PLOS Global Public Health’s publication criteria ? Is the manuscript technically sound, and do the data support the conclusions? The manuscript must describe methodologically and ethically rigorous research with conclusions that are appropriately drawn based on the data presented.

Reviewer #3: Yes

3. Has the statistical analysis been performed appropriately and rigorously?

Reviewer #3: Yes

4. Have the authors made all data underlying the findings in their manuscript fully available (please refer to the Data Availability Statement at the start of the manuscript PDF file)?

Reviewer #3: Yes

5. Is the manuscript presented in an intelligible fashion and written in standard English?

Reviewer #3: Yes

6. Review Comments to the Author

Reviewer #3: Dear Authors,

Thank you for submitting a revised version of your manuscript. This is a significant improvement and can be published in PLOS Global Public Health.

The operationalization of outcome variables is now clearly presented. The conceptual framework upon which the work rests is clear. However, the conceptual/theoretical orientation is hanging. It was only discussed in the beginning part of the work (pages 7-9). The extent to which the assumptions of the CSCFH support/contradict the results of your study needs to be discussed in your results and discussion of findings. Which region has made progress and which has failed as far as child deprivation is concerned? To what extent is this a function of the region's socio-cultural and economic outlook and beliefs?

7. PLOS authors have the option to publish the peer review history of their article (what does this mean? ). If published, this will include your full peer review and any attached files.

**Do you want your identity to be public for this peer review?** For information about this choice, including consent withdrawal, please see our Privacy Policy .

Reviewer #3: No

---

## [Decision Letter · Decision Letter 3]

PGPH-D-24-01755R3

Regional variations in child health deprivation and its associated factors in Nigeria.

Dear Dr. Kupoluyi,

Thank you for submitting your manuscript to PLOS Global Public Health. After careful consideration, we feel that it has merit but does not fully meet PLOS Global Public Health’s publication criteria as it currently stands. Therefore, we invite you to submit a revised version of the manuscript that addresses the points raised during the review process.

We look forward to receiving your revised manuscript.

Kind regards,

Parvati Singh, PhD

Academic Editor

Journal Requirements:

Additional Editor Comments (if provided):

Reviewers' comments:

Reviewer's Responses to Questions

**Comments to the Author**

1. If the authors have adequately addressed your comments raised in a previous round of review and you feel that this manuscript is now acceptable for publication, you may indicate that here to bypass the “Comments to the Author” section, enter your conflict of interest statement in the “Confidential to Editor” section, and submit your "Accept" recommendation.

Reviewer #3: (No Response)

2. Does this manuscript meet PLOS Global Public Health’s publication criteria ? Is the manuscript technically sound, and do the data support the conclusions? The manuscript must describe methodologically and ethically rigorous research with conclusions that are appropriately drawn based on the data presented.

Reviewer #3: Yes

3. Has the statistical analysis been performed appropriately and rigorously?

Reviewer #3: Yes

4. Have the authors made all data underlying the findings in their manuscript fully available (please refer to the Data Availability Statement at the start of the manuscript PDF file)?

Reviewer #3: Yes

5. Is the manuscript presented in an intelligible fashion and written in standard English?

Reviewer #3: Yes

6. Review Comments to the Author

Reviewer #3: Dear author,

Thank you for submitting the revised version of your manuscript. I looked at the different versions of the manuscript, including the original version and revisions 1, 2 and 3. Below are my comments to your previous version:

“The operationalization of outcome variables is now clearly presented. The conceptual framework upon which the work rests is clear. However, the conceptual/theoretical orientation is hanging. It was only discussed in the beginning part of the work (pages 7-9). The extent to which the assumptions of the CSCFH support/contradict the results of your study needs to be discussed in your results and discussion of findings. Which region has made progress and which has failed as far as child deprivation is concerned? To what extent is this a function of the region's socio-cultural and economic outlook and beliefs?”

That was my honest and sincere evaluation of your revised manuscript. I acknowledged that the theoretical framework was clear. All you needed to do was to integrate it into the discussion section. However, you have completely removed the section on theoretical framework titled “Rationale for investigating regional variations” and only discuss the regional issues briefly. This makes the work unscholarly. Now, CSCFH has been stated without reference to the origin of the theory or earlier works therein. I am not sure if this was deliberate or an oversight. The R2 version of the manuscript read even better than the current version.

I understand that this is your manuscript, but I must sincerely say that the current version is unscholarly. The title of your work is “Regional variations in child health deprivation and its associated factors in Nigeria.” That particular section on “rationale for investigating regional variations in Nigeria” gave theoretical and conceptual weight to your study, but you have removed it and made it look like CSCFH is original to you. This is not good scholarship!

I see you have integrated the CSCFH into the discussion, but what sense does it make now when the section on theoretical framework has been removed and only briefly mentioned as part of the introduction? You need to bring back the theoretical framework as it was in the previous version PGPH-D-24-01755R2 and leave the current discussion as is.

7. PLOS authors have the option to publish the peer review history of their article (what does this mean? ). If published, this will include your full peer review and any attached files.

**Do you want your identity to be public for this peer review?** For information about this choice, including consent withdrawal, please see our Privacy Policy .

Reviewer #3: No

---

## [Editor Report · Decision Letter 4]

Regional variations in child health deprivation and its associated factors in Nigeria.

PGPH-D-24-01755R4

Dear Dr. Kupoluyi,

We are pleased to inform you that your manuscript 'Regional variations in child health deprivation and its associated factors in Nigeria.' has been provisionally accepted for publication in PLOS Global Public Health.

Best regards,

Parvati Singh, PhD

Academic Editor